# An Evaluation of Executive Control Function and Its Relationship with Driving Performance

**DOI:** 10.3390/s21051763

**Published:** 2021-03-04

**Authors:** Lirong Yan, Tiantian Wen, Jiawen Zhang, Le Chang, Yi Wang, Mutian Liu, Changhao Ding, Fuwu Yan

**Affiliations:** 1Foshan Xianhu Laboratory of the Advanced Energy Science and Technology Guangdong Laboratory, Foshan 528200, China; lirong.yan@whut.edu.cn (L.Y.); tiant_wen@163.com (T.W.); wen1160281134@163.com (J.Z.); 2Hubei Key Laboratory of Advanced Technology for Automotive Components, Wuhan University of Technology, Wuhan 430070, China; cl455508373@163.com (L.C.); wangyi_echo2020@163.com (Y.W.); liumutian0828@163.com (M.L.); teisyoukou@nsurg.med.osaka-u.ac.jp (C.D.); 3Hubei Collaborative Innovation Center for Automotive Components Technology, Wuhan 430070, China; 4Hubei Research Center for New Energy & Intelligent Connected Vehicle, Wuhan University of Technology, Wuhan 430070, China

**Keywords:** attention, executive control, simulated driving, task-cuing experiment, electroencephalogram, fronto-parietal network

## Abstract

The driver’s attentional state is a significant human factor in traffic safety. The executive control process is a crucial sub-function of attention. To explore the relationship between the driver’s driving performance and executive control function, a total of 35 healthy subjects were invited to take part in a simulated driving experiment and a task-cuing experiment. The subjects were divided into three groups according to their driving performance (aberrant driving behaviors, including lapses and errors) by the clustering method. Then the performance efficiency and electroencephalogram (EEG) data acquired in the task-cuing experiment were compared among the three groups. The effect of group, task transition types and cue-stimulus intervals (CSIs) were statistically analyzed by using the repeated measures analysis of variance (ANOVA) and the post hoc simple effect analysis. The subjects with lower driving error rates had better executive control efficiency as indicated by the reaction time (RT) and error rate in the task-cuing experiment, which was related with their better capability to allocate the available attentional resources, to express the external stimuli and to process the information in the nervous system, especially the fronto-parietal network. The activation degree of the frontal area fluctuated, and of the parietal area gradually increased along with the increase of CSI, which implied the role of the frontal area in task setting reconstruction and working memory maintaining, and of the parietal area in stimulus–Response (S–R) mapping expression. This research presented evidence of the close relationship between executive control functions and driving performance.

## 1. Introduction

Traffic safety has a great impact on the family and society. The World Health Organization (WHO) reported that approximately 1.25 million people died in road traffic accidents every year [1]. Among the traffic accidents, a very large proportion was caused by the drivers, which was nearly 90% according to the National Motor Vehicle Crash Causation Survey (NMVCCS) [2]. The driver, as the final service object, is the central node of sensation and control in the driver-vehicle-environment system and plays the most important role in traffic safety [3]. Drivers’ physical and psychological state would greatly affect driving safety. The abnormal state of the driver, such as distraction and fatigue, would result in visual disturbances, which were related to most accidents [2]. Driving distraction and fatigues are the ubiquitous problems and the major cause of injury and death for the drivers throughout their life cycle [4]. Driving fatigue, usually resulted from lack of sleep or prolonged driving, would cause a decreased function of the sensory-motion system and a decline in driver’s attention ability [5]. Driving distraction is defined as any activity that detracts the driver from the primary driving task, and is mainly reflected in three aspects, i.e., visual (taking one’s eyes off the road), manual (taking one’s hand off the wheel) and cognitive (taking one’s mind away from the driving task) distraction [6]. Both driving fatigue and distraction are the manifestations of insufficient attention allocated for the driving tasks.

Several studies have investigated the monitoring method of the driving attentional state. Generally, several kinds of methods were developed, based on either the behaviors, the psychophysiological state of the driver, or the driving parameters of the vehicle. Some behaviors of the drivers, such as nodding, yawning and mouth movements were closely related to fatigue [7]. Usually, these behaviors were recorded and then analyzed to extract the fatigue-related features, such as Percentage of Eyelid Closure over the Pupil (PERCLOS) [8], the manipulation of the steering wheel [9], etc. Some studies demonstrated the correlation of the physiological parameters of the driver with driving attention, such as the high-frequency electrocardiogram (ECG) component [10] and the EEG (electroencephalogram) signals of the frontal areas [11]. The trajectory and the state of the vehicle, such as the speed, acceleration, and driving direction can also be utilized for distraction detection [12]. Studying the mechanism and the influential factors of attention can help to accurately evaluate the driver’s alert state, replace the passive safety control strategy by active monitoring, improve the driving safety and effectively reduce the occurrence of traffic accidents.

The cognitive studies on attention included the behavioral [13], psychological [14,15], and neuroimaging schemas [16,17] both in subjects with attention-related disorders such as attention deficit hyperactivity disorder (ADHD) [18] and in normal people. Attention is characterized as the ability to effectively block outside distractions while focusing on a single object or task, which is a general function of the whole brain. The neuroimaging studies indicated that several neural networks were involved in attentional functions [19], among which three subsystems were specifically conceptualized, which were alerting, orienting and executive control [20]. Alerting is defined as reaching and maintaining a state that is highly sensitive to incoming stimuli, which would activate the anterior attention system, including the frontal cortex, posterior parietal cortex, and thalamus [20]. Alerting subsystem maintains the alert state and acts on the posterior attention system to support visual orienting. The orienting subsystem screens information from alert input to divert attention to the selected or focused stimulus, which is related with the activities of the frontal eye field, superior parietal cortex, temporal parietal junction, frontal eye fields, and superior colliculus [21]. The executive control subsystem monitors and resolves conflicts between thoughts, feelings, and responses, and plays a crucial role in attention, decision-making and complex conflict processing [20]. Currently, the most used paradigm to study the executive control function is the task-cuing paradigm. In this paradigm, the subjects would perform two or more types of tasks randomly under the instruction of a cue, which would be presented before or at the same time each target appears and prompt the type of task to be performed. The performance efficiency, such as RT and error rate, and the neuroimaging indexes, such as the EEG signal and the functional magnetic resonance imaging (fMRI) signal [19,22], would be recorded and compared between task switching and task repetition conditions. Results indicated that the response was slower, and the error rate was usually higher under the task switching condition, which was called the switch cost. Switch cost is an important indicator to quantify the function of executive control. Theoretical accounts of executive control assumed that multiple components were involved in activating a task-set, including paying attention to new cue-task connections, inhibiting the expression of previous task setting rules, shifting attention to relevant stimulus attributes, activating a goal representation, reconstructing the task’s S–R (stimulus response) rules, setting response criteria and store task settings in working memory [23,24,25,26,27]. The switch cost was believed to occur during the active task setting reconstruction process. Better capability of the task setting reconstruction and complex cognitive processes optimization would result in the reduction of switching cost, which implied the higher efficiency of executive control function in cognitive processes coordination [25,26,27]. The switch cost, to some degree, is the behavioral manifestation of the executive control function. The spatiotemporal activities of the brain, on the other hand, laid the psychophysiological foundation of the executive control function. Several brain areas, including the prefrontal cortex, temporal cortex and anterior cingulate gyrus [18,22,28], were involved. Their activities varied among people with different attentional states, such as stronger activation of the dorsal anterior cingulate cortex, middle temporal gyrus, precuneus, lingual gyrus, precentral gyrus and insula in ADHD patients compared with the healthy adults under the task switching condition [18]. Besides, the psychological experiments demonstrated that the attentional state and CSIs were closely related. For example, the switch cost would increase if the CSI was too short [29,30]. The dynamic relationship between switch cost and brain activities is important to evaluate the executive control function, and is worthy of further research.

The executive control functions should be closely related to the driving performance. To test this hypothesis, quantitatively analyze the behavioral manifestations of the executive control functions, and explore the underlying cognitive mechanism, a total of 35 subjects were recruited to participate in a simulated driving experiment and a task-cuing experiment. The dataset including their driving behavior and EEG signals were acquired. The subjects were divided into three groups according to their driving performance (aberrant driving behaviors, namely lapses and errors). The performance efficiency and brain activation characteristics under different task transition types and CSI levels in different groups were analyzed. The results demonstrated the close relationship between driving performance and executive control efficiency. The fronto-parietal network participated in the executive control process and had a specific function in task setting construction and working memory maintenance.

## 2. Materials and Methods

### 2.1. Method Overview

The main research work was organized as follows: (i) simulated driving experiment and task-cuing experiment; (ii) systematic clustering (SPSS20.0, United States) to divide the subjects into different groups based on the driving performance; (iii) three-way repeated measures ANOVA for behavioral and EEG data among different groups of subjects; (iv) one-way repeated measures ANOVA and paired *T*-test to analyze differences between CSI and task transition types under different groups; (v) one-way ANOVA and two independent sample *T*-test to test the differences among different groups.

### 2.2. Subjects and Experiment Design

A total of 35 right-handed healthy adults (26 males and 9 females; 4 undergraduates, 28 postgraduates, 2 PhD candidates and 1 PhD) with no history of neurological disease were recruited, ranging in age from 21 to 46 (24.9 ± 5.7) years. Their visions were normal or corrected normal. All subjects had a Chinese C1 type (small car) driver’s license with 1 to 17 (3.7 ± 3.1) driving years. They signed the written informed consent. The research was granted by the ethical review committee of Wuhan University of Technology. All subjects participated in two experiments: the simulated driving and the task-cuing experiment.

The simulated driving platform was built by Unity3D (Unity Technologies, Austin, TX, USA) and the Logitech G29 driving simulator (Logitech, Zurich, Switzerland), as shown in Figure 1a. The simulated driving scenario was an approximately 7 km circular orbital road, including slopes, turns, bridge holes, and other elements. Subjects were instructed to sit comfortably wearing the 64-channel Ag/AgCl electrode EEG cap (actiCHamp, Brain Products GmbH, Gilching, Germany), focus on driving along the road, and perform the operation of twisting the steering wheel or braking. The electrodeposition of the EEG electrode cap is shown in Figure 1b. Before the experiment, all subjects had enough time (15 min or so) to familiarize themselves with the driving scene, brake pedal, acceleration torque, and steering wheel sensitivity to prepare for the experiment. During the driving process, each subject was required to complete three driving tasks at a speed limit of 70 km per hour, and each driving task included four laps. After each task, the participants took a short break of five minutes to avoid driving fatigue. The Logitech G29 provided similar force feedback of the steering wheel and brake as real driving. No subjects reported discomfort or driving sickness.

The task-cuing experiment was designed by E-Prime3.0 (Psychology Software Tools Inc., Sharpsburg, PA, USA) and presented on a 19-inch liquid crystal display (LCD) monitor with a screen resolution of 1600*900 (Figure 2). The task-cue was a white picture of a circle or triangle (6 cm × 6 cm) in a black background. The stimulus was a random number from 1 to 9 (except for 5) in red or green. The subjects sat in front of the screen with their sightline on the screen center, wore the EEG cap (actiCHamp, Brain Products GmbH, Gilching, Germany), and were instructed to respond to two types of tasks according to the task-cue. Task A: If the task-cue was a triangle, the subjects needed to judge the color of the number, and press “1” for red or “2” for green. Task B: If the task-cue was a circle, the subjects needed to judge the size of the number, and press “1” for numbers smaller than 5 or “2” for bigger than 5. There is also a task transition type that needed to be reminded about the trials. The task was either repeated or switched relative to the previous trial. According to the execution instructions of the task-cue, the participants were required to distinguish the color or size of the number. If the current task was different from the previous one, the current trial was classified as a switching trial; if the current task was the same as the previous one, the current trial was classified as a repeat trial. This factor was checked to see whether or not the switching trial has an impact on executive control over the repeat trial.

All subjects conducted seven sessions of the task-cuing experiment. Each session contained 42 trials, in which two kinds of tasks appeared randomly and evenly. The occurrence of different events in the same task was different, which was 2:1 of red to green ratio, and 2:1 of bigger-than-5-number to smaller-than-5-number ratio. In each trial, a “+” was shown for 100 ms, then an empty screen for 250 ms, followed by the cue for 100 ms and then the stimulus. The CSI between the cue and the stimulus was set at seven levels, i.e., 200 ms, 400 ms, 600 ms, 800 ms, 1000 ms, 1200 ms, and 1400 ms, which distributed randomly and evenly in each session. The stimulus would not disappear until the subjects pushed a button. After the reaction of the subjects, an empty screen would be shown for 500 ms.

All subjects practiced before the formal experiment to get familiar with the task protocols. During the experiment, they could take a short break between two sessions.

### 2.3. Data Acquisition

In the simulated driving experiment, the driving data and EEG data were recorded simultaneously. The driving data, including the vehicle position and the steering wheel rotation angle, were acquired by the C# scripts based on Unity3D. The EEG data was collected at 1000 Hz by the Biopac actiCHamp Amplifier and BrainVision PyCorder (Brain Products GmbH, Gilching, Germany). The cap worn by the subjects was referenced to the FCz electrode according to the international 10–20 system protocol. The whole driving process of the vehicle on the screen was recorded by Apowersoft (Apowersoft, Hong Kong, China). For the task-cuing experiment, the behavioral data, including the RTs and error rates of the subjects were recorded by the E-DataAid module of E-Prime. The task transition type of each trial except the first one was defined as either repeated or switched relative to the previous trial, i.e., task repetition or switching.

### 2.4. Analysis of Behavioral Data

The driving performance of the subjects was evaluated according to the recorded screen video in the driving process. Specifically, the errors (severe accidents of driving out of the road or car collisions in which situation the vehicle was out of control and needed to be reset to the normal state by the experimenter) and lapses (moderate accidents resulted in off-road but under-control vehicle) made during the simulated driving experiment were counted. Systematic clustering was applied to these two types of errors to divide the subjects into different groups.

The behavioral data in the task-cuing experiment, including the RTs and the error rates under different conditions (group category, task transition type, and CSI) were analyzed. The differences in task activation among the subjects were tested using a 3 (group category: group 1, group 2, group 3) × 2 (task transition type: task repetition, task switch) × 7 (CSI: 200 ms, 400 ms, 600 ms, 800 ms, 1000 ms, 1200 ms, 1400 ms) repeated measures ANOVA (SPSS20.0, United States).

### 2.5. Analysis of EEG Data

The preprocessing of the EEG data was carried out using the EEGLAB toolbox (Swartz Center for Computational Neuroscience, San Diego, CA, USA) in MATLAB (R2013b, MathWorks, Natick, MA, USA). The signal in Fp1 and Fp2 channels were removed from the subsequent statistical analysis due to the disturbance of the electrooculogram (EOG). TP9 and TP10 were selected as the re-reference electrodes. Bandpass filtering (0.1–35 Hz) was applied to remove the noise. By extracting data epochs (200 milliseconds before stimulation to 1500 ms after stimulation) from the continuous EEG signal and data averaging, event information was obtained and event-related potential (ERP) images were created. Finally, independent component analysis (ICA) was applied to remove eye artifacts (including the signal artifacts due to the movement of the eyeball, ocular muscles, and eyelid), ECG artifacts, electromyography (EMG) artifacts, and other noises.

By behavioral data analysis, the EEG data were also analyzed by 3 × 2 × 7 repeated measures ANOVA. The *F* values in the analysis result of variance were extracted to draw the topographic maps, the interactions and single-factor effects were analyzed. Paired *T*-test (testing for activation differences between different task transition types), one-way repeated measures ANOVA (testing for activation differences under different CSI conditions), one-way ANOVA (test the differences among three groups) and two independent sample *T*-test (testing for activation differences between any two groups) were used to explore the effects of various factors on the implementation of executive control mechanisms.

## 3. Results

### 3.1. Behavioral and EEG Characteristics of Different Groups of Subjects

#### 3.1.1. Grouping Results

The 35 subjects were divided into different groups according to their aberrant behaviors (errors and lapses) using the systematic clustering (“bottom-up” aggregation, Euclidean distances, shortest distance algorithm). Initially each subject belonged to the different categories. Then, the pair of subjects with the shortest distance were merged into one category. The distance between this category and the other categories were calculated and merged the two nearest categories. Continue this procedure until all the categories were merged into one. Three categories were set in advance and the subjects were classified according to the pedigree cluster diagram. The clustering results based on the driving data are shown in Figure 3. Subject 7, 17, 21, 23, 27 and 28 were classified as group 1, subject 1, 8, 10, 14, 15, 16, 19, 24, 25, 26, 29, 30, 32, 33, and 34 were classified as group 2, and the rest were classified as group 3. There was no significant difference in genders, ages, driving years and education levels among the three groups (*χ*^2^ = 4.836, *P* = 0.089; *F* = 0.149, *P* = 0.862; *F* = 0.102, *P* = 0.903; *χ*^2^ = 2.978, *P* = 0.561 respectively). The average numbers of errors, lapses and all aberrant driving behaviors (summed numbers of lapses and errors) in group 1, group 2 and group 3 were (7.67 ± 5.75, 20.5 ± 1.61, 28.17 ± 6.85), (9.73 ± 3.83, 10.87 ± 2.29, 20.6 ± 4.32) and (2.93 ± 2.58, 2.86 ± 1.75, 5.79 ± 3.51) respectively. The mean occurrence of the errors, lapses and total occurrence in the three groups were significantly different (*F* = 12.152, 170.065 and 64.951 respectively). The post hoc pair-wise comparison indicated significant difference of lapses (*T* = 9.202, 20.644, and 10.515), and all the aberrant driving behaviors (*T* = 3.065 of group 1 vs. group 2, 9.789 of group 1 vs. group 3, and 10.084 of group 2 vs. group 3, all *P* < 0.01). The occurrence of errors was significantly different between group 2 and group 3 (*T* = 5.709, *P* < 0.01). The difference of the errors in group 1 vs. group 2 and group 3 was not significant (*T* = −0.969, *P* = 0.345; *T* = 1.941, *P* = 0.102 respectively).

#### 3.1.2. Effect of Task Transition Types, CSIs and Group on the Behavioral Data

The three-way repeated measures ANOVA on RTs (reaction times) analysis indicated that the main effects of task transition type and CSI on RTs were significant (*F* (1, 32) = 35.531, *P* = 0.000, and *F* (6, 192) = 7.769, *P* = 0.000 respectively). The main effect of group (*F* (2, 32) = 2.986, *P* = 0.065), the threesome interaction effect (*F* (12, 192) = 1.532, *P* = 0.115) and pair-wise interaction effects (*F* (6, 192) = 1.541, *P* = 0.167; *F* (2, 32) = 1.233, *P* = 0.305; *F* (12, 192) = 0.772, *P* = 0.679) were not significant. Generally, in the three groups, the RTs were basically smaller under the task repetition condition than those under the task switching condition (Figure 4). The mean RT of group 1 (Figure 4a) was 799 ms and 909 ms for the task repetition and task switching condition respectively, of group 2 (Figure 4c) was 866 ms and 980 ms respectively, and of group 3 (Figure 4e) was 744 ms and 809 ms respectively. The RT in group 1 was shorter than that of group 2 but longer than that of group 3 (not significant). In group 1, when the CSI was lower than 800 ms, the switch cost fluctuated around 150 ms to 200 ms, as the CSI continued to increase, the switch cost first decreased and then increased, reaching the minimum when the CSI was 1200 ms. In group 2, the switch cost first increased along with the increasing of CSI and then fluctuated around 100 ms. In group 3, the switch cost fluctuated around 70 ms and reached minimum when the CSI was 400 ms. The mean switch cost of RT in group 1, group 2 and group 3 was 110 ms, 114 ms and 65 ms respectively.

The main effect of task transition type on the error rates was significant (*F* (1, 32) = 6.154, *P* = 0.019). The main effects of CSI and group (*F* (6, 192) = 0.546, *P* = 0.773; *F* (2, 32) = 2.673, *P* = 0.084), the threesome interaction effect (*F* (12, 192) = 0.661, *P* = 0.787) and the pair-wise interaction effects (*F* (6, 192) = 0.563, *P* = 0.759; *F* (2, 32) = 0.979, *P* = 0.387; *F* (12, 192) = 0.933, *P* = 0.515) were not significant. The mean error rate of group 1 was 1.2% and 1.7% under the task repetition and switching condition respectively; of group 2 was 2.9% and 4.8% respectively, and of group 3 was 2.7% and 2.9% respectively. Generally, for the three groups, the error rates were smaller under the task repetition condition than that under the task switching condition, and the error rate of group 2 was higher than the other two groups. In group 1, the switch cost fluctuated around 0.5%, and the smallest absolute value appeared when the CSI was 200 ms (Figure 4b). The switch cost of group 2 had a downward trend as the CSI increased, except for an abnormal increase when the CSI was 1000 ms (Figure 4d). In group 3, the switch cost generally decreased along with the increasing of CSI, but increased when the CSI was 800 ms and 1400 ms, respectively (Figure 4f). The mean switch cost of error rate in group 1, 2 and 3 was 0.6%, 1.9% and 0.1% respectively.

#### 3.1.3. Effect of Task Transition Types, CSIs and Group on the EEG Data

The main effect of CSI on brain activity was significant in the most frontal and parietal (Figure 5b, all electrodes except TP7, P7, PO7, O1, Oz and Iz), of group in the prefrontal (AFz, AF3, AF4 and AF8), the frontal (F1, F2, F3, F4, F6, F7 and F8), the frontal-central (FC1, FC2, FC4 and FC6), the central (Cz and C1) and the right fronto-temporal regions (FT8, Figure 5c). The main effect of task transition type (Figure 5a), the threesome interaction effect (Figure 5d) among group, task transition type and CSI was not significant. The pairwise interaction effects were significant at several limited electrodes (AFz, F1, F2, T8 and PO4 in Figure 5e, FC2, Cz and C2 in Figure 5f).

Considering the existence of the interaction effects, the post hoc comparison was performed to test the simple effects of the task transition type, CSI and group.

In group 1 (Figure 6a), the repeat trials caused stronger activation in the left prefrontal region (AF3) when the CSI was 800 ms, and the switching trials caused significantly stronger activation in the right parietal cortex (P8) as the CSI increased to 1400 ms. The difference in CSI was mainly concentrated in the prefrontal (AFz, AF7 and AF8), frontal (F1, F3 and FC1), fronto-temporal (FT7 and FT8) central (C2, C4 and C6) and central-parietal (CP2 and CP4) regions under the task repetition condition, and in the frontal (centered at the F1 electrode), right fronto-temporal (FT8) and right central parietal (centered at the C2 electrode) regions under the task switching condition.

In group 2 (Figure 6b), the repeat trials caused stronger activation in the left central parietal (CP1) and parietal regions (P1) when the CSI was 200 ms, the switching trials caused significantly stronger activation in the right prefrontal region (AF8) when the CSI was 1400 ms. The difference in CSI was mainly concentrated in the frontal (F2, FC2 and FC4), parietal (P6) and parietal-occipital (PO8) regions under the task switching condition, while not significant under the task repetition condition.

In group 3 (Figure 6c), compared to the switch trials, the repeat trials caused significantly stronger activation in the parietal (Pz and P3) and parietal-occipital (POz and PO3) regions when the CSI was 600 ms, and the switching trials caused significantly stronger in the right fronto-temporal (FT8), temporal (T8), temporal parietal (TP8), most central frontal (a large area centered in the FCz electrode) regions when the CSI was 1400 ms. The difference in CSI was mainly concentrated in the frontal (F1, F2, FC1 and FC2), central (Cz, C1, C2 and C4) and central parietal (CPz, CP1, CP2 and CP4) regions under the task repetition condition, and under the task switching condition, in most of the prefrontal, frontal, parietal, temporal and parietal-occipital regions (up to AFz, down to POz, left to FT7, and right to FT8 electrodes), which was the fronto-parietal network.

For the difference among the three groups (Figure 7a), under the task repetition condition, the brain activation differences were in the right frontal (F2, F4, F6, F8 and FC6) and right fronto-temporal (FT8) regions when the CSI was 400 ms, in the right frontal-central (FC6), left fronto-temporal (FT7 and FT9), central (Cz, C1, C2, C3, C4 and C6) and central-parietal (CP1) regions when the CSI was 1000 ms, in the frontal-central (FC2), right fronto-temporal (FT8) and central (Cz) regions when the CSI was 1200 ms, and in most of the prefrontal and central regions (centered at the F1 electrode) when the CSI was 1400 ms. Under the task switching condition, the brain activation differences were in the right parietal-occipital (PO8) region when the CSI was 400 ms, in the prefrontal (AFz and AF4), frontal (F4), right parietal-occipital (PO8) and occipital (O2 and Iz) regions when the CSI was 600 ms, in the prefrontal (AFz), right frontal (F2, F4, F6 and FC4), parietal (CP1, P1 and P5), left parietal-occipital (PO7) and occipital (O1) regions when the CSI was 800 ms, in the prefrontal (AFz), frontal (F1, F2, F3, F5, FC2 and FC5), left fronto-temporal (FT7) and central (Cz, C1 and C6) regions when the CSI was 1000 ms, in the frontal and central parietal regions (part of the area centered at the Cz electrode) when the CSI was 1200 ms. When the CSI increased to 1400 ms, the brain activation differences occurred in almost all areas of the prefrontal, frontal, bilateral temporal and central parietal regions.

For the difference between group 1 and group 2 (Figure 7b), the brain activations in group 1 were more intense, in the frontal (centered at the F2 electrode, CSI = 400 ms), central parietal (CPz and CP1, CSI = 600 ms) and frontal-parietal (most areas of the frontal and parietal regions, CSI > 800 ms) regions under task repetition condition, in the frontal (centered at the F2 electrode, CSI = 600 ms), parietal (centered at the CP1 electrode, CSI = 800 ms and 1400 ms) and frontal-parietal (most areas of the frontal and parietal regions, CSI = 1000 ms and 1200 ms) regions under task switching condition. The brain activations in group 2 were stronger in the right parietal-occipital (PO8, CSI = 400 ms, 600 ms, 800 ms and 1000 ms) and occipital (Iz, CSI = 600 ms) regions.

For the difference between group 1 and group 3 (Figure 7c), the brain activations in group 1 were stronger, in the right central (C2, C4, FC4 and FC6, CSI = 400 ms) and bilateral fronto-temporal (FT7 and FT8, CSI = 400 ms and 1000 ms) regions under task repetition condition, in the parietal-occipital (centered at the PO3 electrode, CSI = 800 ms) and right fronto-temporal (FT8, CSI = 1200 ms) regions under task switching condition. The brain activation in group 3 was stronger in the frontal central region (FC2, CSI = 1400 ms) under task switching condition.

As for the difference between group 2 and group 3 (Figure 7d), the brain activation in group 3 was stronger, in the central (FC3, Cz and CP1, CSI = 1000 ms and 1200 ms) and fronto-parietal (most areas of the frontal and parietal regions, CSI = 1400 ms) regions under task repetition condition. Under the task switching condition, the brain activations in group 2 were stronger in the left parietal (P7, CSI = 800 ms), parietal-occipital (PO7 and PO8, CSI = 400 ms, 600 ms and 800 ms) and occipital (Oz, O1, O2 and Iz, CSI < 1000 ms) regions. With the increase of CSI, the brain activation intensity and activation range in group 3 gradually increased, and when the CSI was 1400 ms, the brain activation area almost covered the entire frontal-parietal network.

## 4. Discussion

In this work, a total of 35 healthy subjects were recruited to participate in a simulated driving experiment and a task-cuing experiment. The subjects were divided into three groups according to their driving performance. Then the performance efficiency and EEG data acquired in the task-cuing experiment were compared among the three groups, and the effect of task transition types and CSIs was statistically analyzed. The performance efficiency and the underlying cognitive mechanism of the executive control function, and its relationship with the driving performance was investigated.

### 4.1. Relationship between Driving Performance and Executive Control Efficiency

Driving is a very complicated procedure, which is composed of a series of behavioral operations, and resulted from the dependable perception-decision-execution cycle of the brain. The driving performance can be studied using the number of crashes, the number of incorrect use of turn signals, overtaking distance, vehicle trajectory and speed, etc. [12,31]. Reason et al. [32] presented a useful theoretical model by using the risky driving behaviors for the driving performance evaluation. Particularly three categories of aberrant behaviors related to different cognitive and decisional processes were defined, i.e., errors, lapses and violations. Errors were defined as failures to achieve the intended consequences of planned actions (e.g., braking too quickly on a road with low friction), largely representing information-processing deficits. Lapses were defined as failures of attention or memory (e.g., attempt to drive away from traffic light in third gear), largely representing information-sensory deficits. Violations were defined as deliberate violation of rules or failure to follow safe driving practices (e.g., decide to continue driving at the red light). Enlighted by this definition, we defined the errors and lapses in our work according to the severity of the accidents and the controllability of the vehicle. The severe accidents were caused by a series of mistakes made during the information processing procedure, while the moderate accidents were usually resulted from negligence of the external information. Errors and lapses constructed two dimensions to depict the aberrant driving behaviors in our driving scene. Accordingly, the enrolled 35 subjects were divided into three groups. The driving performance was the best in group 3 with the fewest errors and lapses. Group 1 had the highest occurrence of lapses and medium occurrence of errors, and group 2 had the highest occurrence of errors and medium occurrence of lapses.

Executive control refers to the coordination of multiple tasks to complete complex cognitive control processes. task-cuing experiment is a common paradigm to study the underlying mechanism of executive control function. The subjects needed to perform the same task as the former one (task repetition) or quickly switch to another kind of task (task switching). During the experiment, the subjects would maintain a specific cognitive state and construct a task setting process involving perception, attention, memory, and response [33]. Under the task repetition condition, the subjects only needed to implement previously configured task settings. While under the task switching condition, the subjects needed more effort to complete the configuration of a new kind of task. The executive control demands were greater, due to the working memory requirements to maintain multiple tasks in memory, the inhibition of the previous task, and the activation of the current task [34]. Consequently the subjects’ response was usually slower and the accuracy lower, which was considered as the switch cost phenomenon [30]. The switch cost could be utilized as a quantitative indicator and was positively correlated with the subjects’ executive control efficiency [23,27]. In our work, the task transition type had the significant independent impact on RT and error rate, which was significantly larger under the task switching condition for all the groups. Though group effect was not significant, the average switch costs of RTs and error rates in group 1, 2, and 3 were 110 ms and 0.6%, 114 ms and 1.9%, and 65 ms and 0.1%, respectively (Figure 4), which indicated the best executive control performance of group 3, and the worst of group 2. The behavioral performance of group 3 in the task-cuing experiment revealed that group 3 obviously had better capability to allocate the available attentional resources when the demands for the working memory maintaining former information inhibition, and reconfiguration of the current task was greater. This capability also resulted in better driving performance of group 3, which was highly correlated with their attentional and cognitive states. Although the total number of the abnormal driving behaviors of group 2 was lower than that of group 1, group 2 had the most errors, the largest switch cost, and the worst executive control function. This implied that in the two aberrant driving behaviors, error clearly better reflected the executive control function. This would be further ascertained by the EEG results. CSI had a significantly independent impact on RT (Figure 4). As CSI increased, the performance efficiency (including RT and error rate) was significantly improved. The switch cost of the error rate decreased as the CSI increased, especially in group 3. The impact of CSI on behavioral performance and switch cost proved that the task setting reconstruction process, i.e., the preparation effect for new trials [35], was an important source of switch cost. During the experiment, once the task-cue appeared, the brain began to complete the control conversion process from the initial abstract rule representation to the actual representation [22]. When the CSI was short, this process cannot be well executed due to the pressure and insufficiency of the preparation time, which would result in an unstable characterization of the actual stimulus. Whereas when the CSI was longer, the conversion process would be much smoother, and better performance efficiency could be achieved.

### 4.2. The Underlying Cerebral Network for the Executive Control Function

The brain activities under different task transition conditions reflected how the brain was organized to fulfill the executive control function. As can be seen in Figure 5, generally the main effect of task transition type on the EEG activities was not significant (Figure 5a). However, it had the interaction effect with group factor in some electrodes (Figure 5e). The following post hoc simple effect analysis indicated that group 1 (Figure 6a) had stronger activation in AF3 (CSI = 800 ms), group 2 (Figure 6b) in CP1 and P1 (CSI = 200 ms), and group 3 (Figure 6c) in Pz, P3, POz and PO3 (CSI = 600 ms) under the task repetition condition; while group 1 had stronger activation in P8 (CSI = 1400 ms), group 2 in AF8 (CSI = 1400 ms) and group 3 in most electrodes of the fronto-parietal network (CSI = 1400 ms) under the task switching condition. These results suggested the different activation patterns during the executive control procedure in three groups of subjects. The ANOVA results did reveal the significant main effect of group on the EEG data, specifically in the prefrontal (AFz, AF3, AF4 and AF8), the frontal (F1, F2, F3, F4, F6, F7 and F8), the frontal-central (FC1, FC2, FC4 and FC6), the central (Cz and C1) and the right fronto-temporal regions (FT8, Figure 5c), which constituted the fronto-parietal network [17,29]. In general, the activation levels were stronger and activation ranges in the fronto-parietal network were wider in group 3 under the task switching condition. This was responsible for their better capability to reallocate the attentional resources, which also proved that most of the brain regions of the fronto-parietal network were required to complete the task setting process [33,34,35].

The activation comparison among three groups indicated that their difference was stronger under the task switching condition, when the regions extended from a small area in the frontal and central regions (centered at F2) to most areas of the fronto-parietal network with increased intensity as well (Figure 7a). Under both task switch and task repetition conditions, the activation degree of group 1 and group 3 was significantly stronger than group 2 (Figure 7b,d). The weakest activation intensity of group 2 was responsible for their worst performance of the executive control function and their highest occurrence of the driving errors. The underlying regions of interest for the executive control functions and their activity changes, along with the CSI were further analyzed. As for the comparison between group 1 and 2 (Figure 7b), when the CSI was 200–800 ms, the activation range and intensity varied and the difference was mainly concentrated in the prefrontal and frontal regions (around F2). When the CSI was 1000–1200 ms, the difference was stable in most brain regions including the prefrontal, frontal, central, temporal and superior parietal regions. As for the comparison between group 2 and 3 (Figure 7d), the activation difference was mainly concentrated in the frontal (centered at F2, CSI = 800 ms) and frontal-central regions (centered at the FC1, CSI = 1200ms) under the task switching condition, and occupied most of the fronto-parietal network under both conditions when the CSI was 1400 ms. In general, our results indicated the significant effect of the group factor in the frontal-parietal network. Additionally, the instability of frontal region activation revealed its specific role in executive control. It has been suggested that a superordinate fronto-cingulo-parietal network supporting cognitive control may also underlie a series of distinct executive functions, including attention, signal recognition, behavior strategy formulation, motion control, impulse control, and information feedback recognition [22,28,36]. In the task-cuing experiment, during the period from the end of the previous trial to the end of the next task-cue, there was a process of working memory maintenance of multiple task settings, evaluation and reconstruction of the current task setting [22,37]. When the task to perform was switched, the task settings needed to be updated and the extra work was required to suppress the previous task setting, which resulted in stronger cerebral activities. Consistent with the behavioral results, the better performance efficacy and stronger activation of group 3 indicated their better capability to allocate the available attentional resources, to express the external stimuli, and to process the information in the nervous system, especially the fronto-parietal executive control network.

### 4.3. Effect of CSI Level on the Brain Activities

The main effect of CSI on EEG data was significant. Besides, CSI had the interaction effect with groups in some channels (FC2, Cz and C2 in Figure 5f). The influence of CSI levels on the executive control process can be also observed in the following post hoc simple effect analysis. The among-CSI difference of the brain activation existed in mostly the left prefrontal, frontal, right frontal-parietal and bilateral fronto-temporal regions under task repetition condition and in fronto-parietal network (centered at FC2) under task switching condition in group 1 (Figure 6a), in F2, FC2, FC4, P6 and PO8 under task switching condition in group 2 (Figure 6b), in the frontal and central regions (a small area centered at Cz) under the task repetition condition and in the fronto-parietal network (most areas of frontal, central, bilateral temporal and parietal and parietal-occipital regions) under task switching condition in group 3 (Figure 6c). The simple effect analysis indicated that the activation degree and range of brain regions increased along with the increase of CSI. The brain activation differences among different CSIs were mainly concentrated in the fronto-parietal network, which was most strong in group 3, secondly strong in group 1, and the weakest in group 2 (Figure 6). The results indicated that subjects with better attention status and better executive control efficiency were more sensitive to CSI.

In general, the increase of the CSI is helpful for the activation of the task settings and the more effective conversion among different tasks. It is noted that when CSI was 1000 ms, the intensity and range of the brain activation were significantly increased in all three groups and then remained at a high level (Figure 6). Additionally, 1000 ms seemed to be also a key downtrend point of the switch cost, especially for the RT of group 2 and 3 (Figure 4). Both the performance efficiency and the underlying cognitive process reached an optimal level at this CSI. An appropriate CSI might be helpful for the subjects to maintain the balance of task setting reconstruction and S–R mapping expression. Both the task setting reconstruction and S–R mapping expression relied on the working memory, which was an iterative process including encoding, storage, recognition and recall. When the CSI was relatively short, the task-cue processing and task setting reconstruction was needed to be performed synchronously, there was no time for enough iterations, and resultantly, the accuracy of the executive control function could not be guaranteed. On the other hand, when the CSI was long, a series of iterations could be fulfilled. Besides, according to the experience of the subjects, they might even have time for the rehearsal of the expected task. Under this circumstance, the pre-task setting reconstruction process might have already started before the stimulus, and the working memory load would remain at a high level.

Though there existed differences among the three groups, both frontal and parietal regions were involved. The activities of the frontal cortex fluctuated along with the CSI. It was activated when the CSI was 200 ms, and the activation seemed to be weakened when the CSI increased to 400 ms in group 2 and group 3 (Figure 6b,c). As the CSI continued to increase, the frontal activation increased again. The frontal cortex was responsible for maintaining task settings, regulating and controlling task-related behaviors [16]. When the CSI was short, the subjects did not have enough time to classify the stimulus and complete the task setting reconstruction process [38]. The time pressure and the conflict between task setting and S–R mapping required the high degree of participation of the frontal region [29]. When the preparation time is sufficient, the time pressure was reduced, the conflict between task setting and S–R mapping expression was reduced, and the requirement of the cognitive control was also decreased. As a result, the activity of the frontal cortex was weakened. When the CSI increased further, the activation of the frontal cortex was restrengthened, which was related to the increased load of the working memory due to the prolonged time of S–R mapping expression [29,34]. This can also explain the phenomenon that the activity of the parietal cortex, which was sensitive to the conflict of S–R mapping [39], was stronger when the CSI was larger. These results indicated the possible role of the frontal area in task setting reconstruction and working memory maintaining, and of the parietal area in S–R mapping expression.

### 4.4. Novelty and Limitations

In this research, a unified experimental and analytical schema for multimodal data including the behavior, EEG activity and psychological performance was presented to explore the underlying cognitive mechanism of attention in driving. Through the comparative analysis of different groups of participants, the quantitative correlation between executive control function and driving performance was established, and the spatiotemporal activity of the brain during this procedure was revealed. The relationship between dangerous driving behavior and attention, along with CSIs and other parameters, was disclosed. Based on the presented methods and the acquired results, the attentional state of the driver could be monitored through EEG signals to avoid the distraction, and the dangerous driving behaviors including errors and elapses could be prevented. Besides, the attentional and behavioral characteristics of the drivers can be analyzed in advance and the subject-specific driving style can be evaluated. Accordingly, different real-time online human-computer interaction schemes can be provided for different kinds of drivers. Furthermore, the individual’s driving performance and their EEG performance could be mutually corroborated, which would supply new reference for driving training and administration. In general, the presented schema supplies a new kind of intelligent human-computer interaction method, and this active safety control would significantly improve the driving safety. Except for driving, the research findings would be applied to other life risk activities.

The present study is limited principally by the unbalanced gender proportion, uneven ages and driving ages of the subjects. Age, gender and educational background are all the crucial factors affecting the executive control functions, performance efficiency, and brain activities of the human [40,41]. A total of 35 subjects were studied and there are only 6 subjects in group 1. Though the meaningful results were found and no significant difference was detected for ages, genders, driving ages and education backgrounds among the three groups, these results need to be replicated in much larger sample size and the general population. Besides these factors, the other demographic factors of different groups, such as the driving experience including the driving frequency and the load, might all have significant effect on the cognition and behavior of the drivers. The definition of all the related parameters and a larger sample size would be crucial to help consolidate a bigger picture of our work. The individual’s driving performance and their EEG performance could be mutually corroborated, which would supply new reference for driving training and administration. However, in our work, their interaction dynamics cannot be analyzed because of the insufficient repeated measures of the subjects from both the cognitive and the behavioral sides. We would like to conduct a longitudinal cohort study and we believe that very interesting and more robust results would be obtained. Second, the spatiotemporal characteristics of the underlying brain function need to be further studied. In our work, we analyzed the brain electrical activity mapping. As EEG is a kind of scalp electrical signal and has a limited spatial resolution, the location of the anatomical areas might not be very accurate. We observed the participation of the frontal and parietal regions in the executive control process. However, deeper areas in the brain, such as the cingulate gyrus, which has been demonstrated to be a critical part of the fronto-cingulo-parietal network for executive functions [28], cannot be located. The EEG source localization technique might be helpful. The causal relationship among the regions and their dynamic activity can be further analyzed by using the time series analysis methods, such as dynamic causal modeling [42]. Finally, the executive control function is one of the three sub-functions (altering, orienting and executive control) of attention. According to our understanding, alerting and orienting subsystems acted mainly in the information perception level. Compared with them, the executive control subsystem acted mainly in the higher decision and control level. The executive control function would have a direct relationship with the behaviors, such as the switch cost of reaction time and error rate in the psychological experiment and driving performance in the simulated driving experiment. Hence in this work, we focused on the executive control function and found its positive correlation with these behavioral performances. However, the other two sub-functions are also important for the whole process, and their implication in the driving performance is worthy of study. Till now, the relationship among these sub-functions and the underlying mechanism of attention is not yet clear [43]. This warrants further synthetic research of the sub-functions of attention. The driving performance of the drivers was evaluated based on two types of aberrant driving behaviors, i.e., errors and lapses. Although this definition method was relatively common in the research of human factors in engineering and driving behavior [31,32], it was still a subjective judgment method. The objective data such as steering wheel angle and driving route was planned to define abnormal driving behavior in the future research.

## 5. Conclusions

In this work, the simulated driving and task-cuing experiments were conducted, and the correlation between driving performance and executive control function was analyzed. The subjects with lower driving error rates had better performance efficiency as indicated by the RT and error rate in the task-cuing experiment, which was related with their better capability to allocate the available attentional resources, to express the external stimuli and to process the information in the nervous system, especially the fronto-parietal executive control network. The activation degree of the frontal area fluctuated, and of the parietal area gradually increased along with the increase of CSI, which implied the possible role of the frontal area in task setting reconstruction and working memory maintaining, and of the parietal area in S–R mapping expression. This research provided evidence of a close relationship between executive control functions and driving performance, which supplies new reference for intelligent human-computer interaction and active safety control in driving.

## Figures and Tables

**Figure 1 sensors-21-01763-f001:**
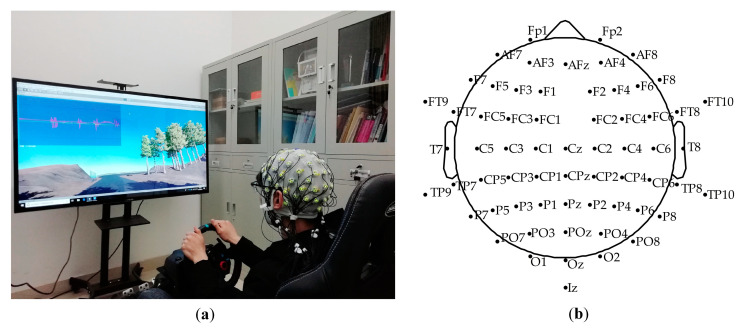
(**a**) Simulated driving platform; (**b**) EEG (electroencephalogram) cap electrode location map.

**Figure 2 sensors-21-01763-f002:**
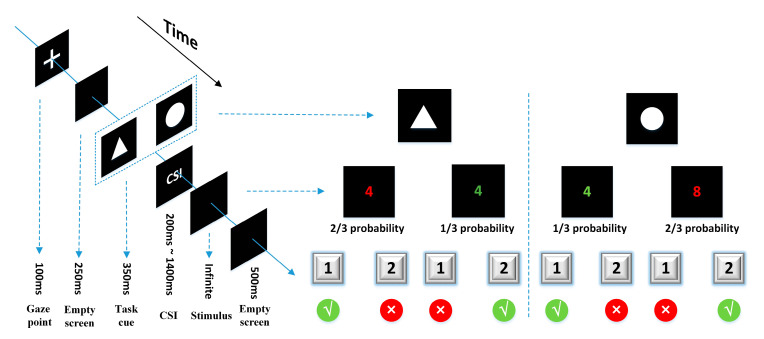
A single trial presentation process and task operation rules of the task-cuing experiment.

**Figure 3 sensors-21-01763-f003:**
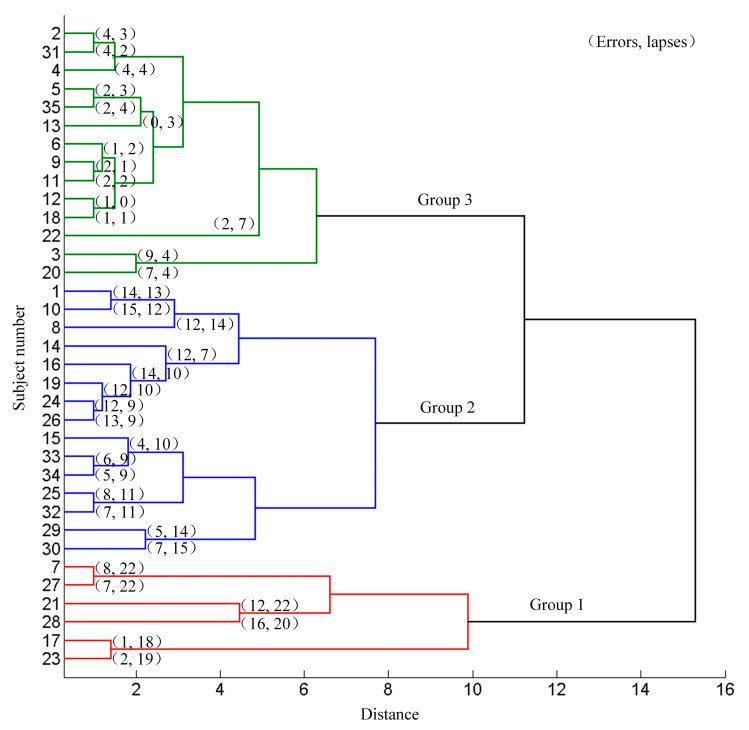
Results of systematic clustering based on the driving data.

**Figure 4 sensors-21-01763-f004:**
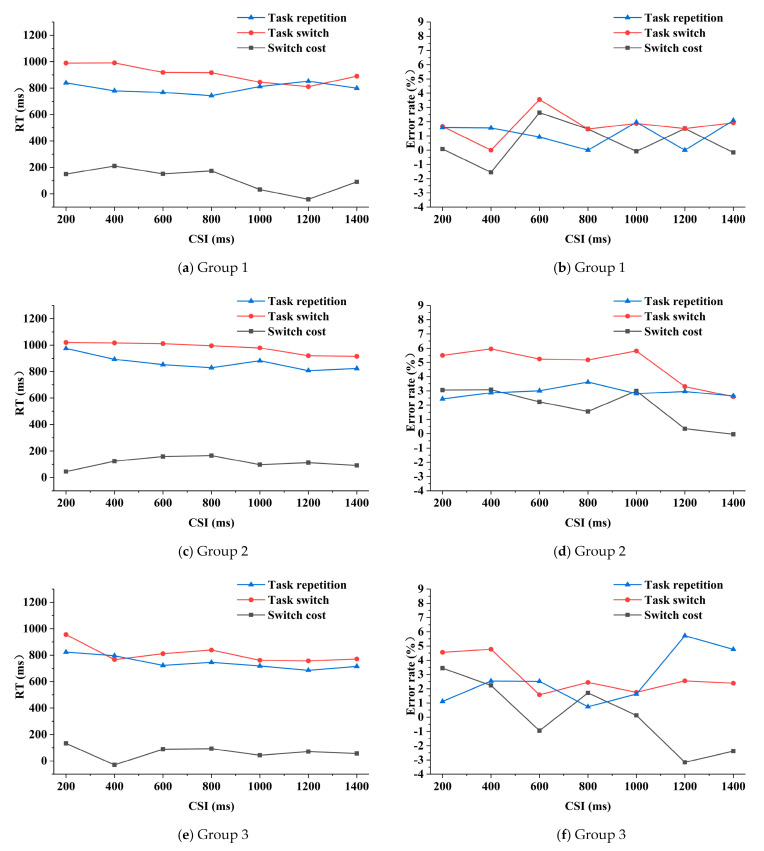
RT (reaction time) and error rate of the three groups as functions of task transition type and CSI (cue-stimulus interval).

**Figure 5 sensors-21-01763-f005:**
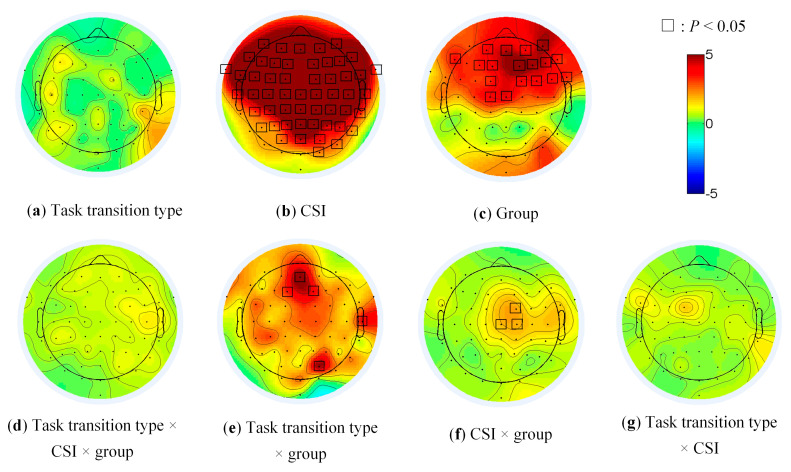
The main effects and interaction of group, task transition type and CSI in EEG data.

**Figure 6 sensors-21-01763-f006:**
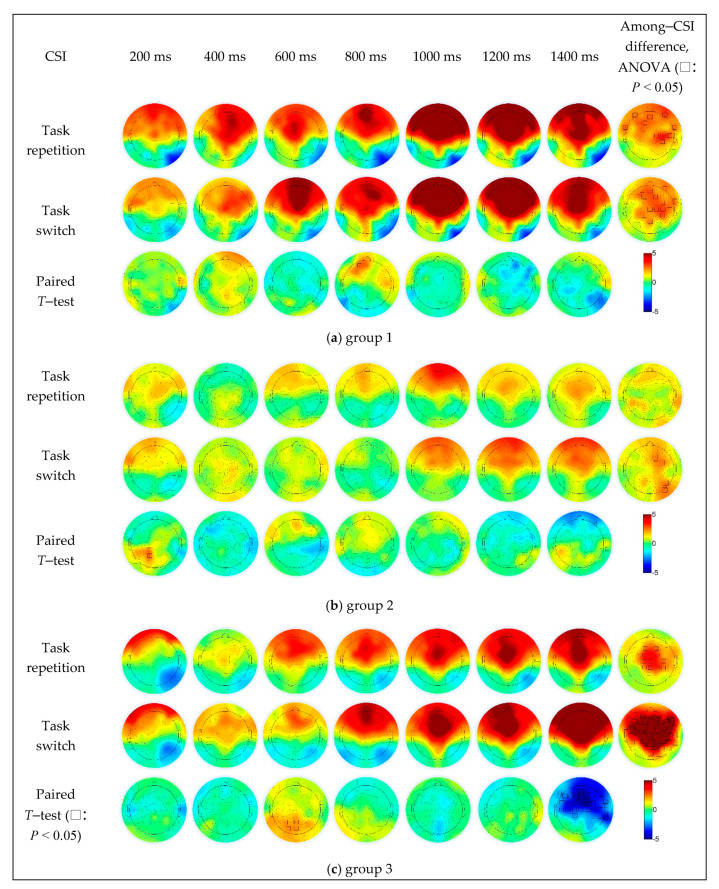
The simple effect of CSI and task transition types on EEG data in different groups.

**Figure 7 sensors-21-01763-f007:**
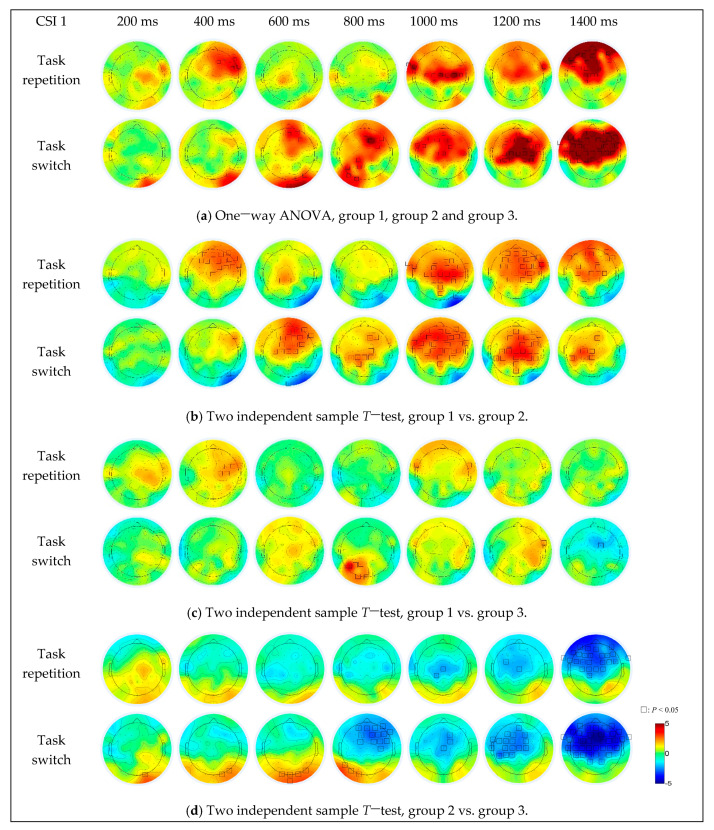
The simple effect of group on EEG data under different CSI conditions and task transition types.

## Data Availability

The data presented in this study are available on request from the corresponding author.

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
