# Peer review of "An Evaluation of Executive Control Function and Its Relationship with Driving Performance"

_sensors, 2021, doi:10.3390/s21051763_

Round 1

Reviewer 1 Report

The present article is well-established and the subject is quite interesting, but some minor revision should be considered.

1. Executive Control Function is a sub-function implied in the process.  The other two sub-functions are important for the whole process. What is the implication of the ECF in the driving performance and can you oversee the implication of the other sub functions?

2. More information about the sample is needed. In particular, what was their education level or other things important for the research?

3. Lines 14-15 contain a phrase 'The executive control process is a crucial sub-function of attention'. What makes it crucial/ that important/ unique? Why are not the other two equally important?

4. The article innovation should be presented in the Introduction.

5. Is there an applicability of the research findings?

6. In the Discussion it would be better to have seen more use of terms like 'originality' and 'significance'. There is no clear conclusion on why the research findings are significant.

Author Response

Response to Reviewer 1 Comments

Comment 1: Executive Control Function is a sub-function implied in the process. The other two sub-functions are important for the whole process. What is the implication of the ECF in the driving performance and can you oversee the implication of the other sub functions?

Response 1: Thanks indeed for your careful reading and insightful questions. According to Posner’s attention network model, the human attention system included three sub-components, i.e. alerting, orienting and executive control [1]. Alerting subsystem maintains the alert state and acts on the posterior attention system to support visual orienting [2]. The orienting subsystem screens information from alert input to divert attention to the selected or focused stimulus [3]. And the executive control subsystem monitors and resolves conflicts between thoughts, feelings and responses in attention [2]. The three subsystems are closely interrelated, and as you pointed out, executive control function is a sub-function implied in the process, and the other two sub-functions are important for the whole process. Till now, the relationship among the three sub-functions is not yet clear [4]. This warrants further synthetic research.

According to our understanding, alerting and orienting subsystems acted mainly in the information perception level. Compared with them, the executive control subsystem acted mainly in the higher decision and control level. The executive control function would have a direct relationship with the behaviors, such as the switch cost of reaction time and error rate in the psychological experiment and driving performance in the simulated driving experiment in our ms. Driving is a very complicated procedure, which is composed of a series of behavioral operations, and resulted from the dependable perception-decision-execution cycle of the brain. Hence in this work, we focused on the executive control function and found its positive correlation with the driving performance. This relationship was ascertained from both the behaviors and cerebral activities. Your comments about the other sub functions and their implication in the driving performance are very constructive. In fact, we have set up the experimental procedures for these sub-functions and acquired several samples. We plan to conduct joint analysis of the three sub-functions and driving performance, and adopt Dynamic Causal Modeling (DCM) to characterize the temporal and spatial activities of the brain in the driving process. We hope that a further work would enhance our understanding about the role of attention in driving.

The manuscript was revised accordingly (4.4. Limitations of this research, Page 17, Line 612) as follows.

According to our understanding, alerting and orienting subsystems acted mainly in the information perception level. Compared with them, the executive control subsystem acted mainly in the higher decision and control level. The executive control function would have a direct relationship with the behaviors, such as the switch cost of reaction time and error rate in the psychological experiment and driving performance in the simulated driving experiment. Hence in this work, we focused on the executive control function and found its positive correlation with these behavioral performances. However, the other two sub-functions are also important for the whole process, and their implication in the driving performance is worthy of study.

Comment 2: More information about the sample is needed. In particular, what was their education level or other things important for the research?

Response 2: Thank for your suggestion. The manuscript was revised accordingly (Page 3, Line 137; Page 6, Line 252; Page 16, Line 589) as follows.

A total of 35 right-handed healthy adults (26 males and 9 females; 4 undergraduates, 28 postgraduates, 2 PhD candidates and 1 PhD) with no history of neurological disease were recruited, ranging in age from 21 to 46 (24.9 ± 5.7) years.

There was no significant difference in genders, ages, driving years and education levels among the three groups (χ2 = 4.836, P = 0.089; F = 0.149, P = 0.862; F = 0.102, P = 0.903; χ2 = 2.978, P = 0.561 respectively).

Though the meaningful results were found and no significant difference was detected for ages, genders, driving ages and education backgrounds among three groups, these results need to be replicated in much larger sample size and the general population.

Comment 3: Lines 14-15 contain a phrase 'The executive control process is a crucial sub-function of attention'. What makes it crucial/ that important/ unique? Why are not the other two equally important?

Response 3: Your questions are very interesting and suggestive. Among the three sub-functions of attention, alerting refers to reaching or maintaining a state in which the sensitivity to upcoming information increases, orienting refers to the shifting of attention to the stimulus to be selected or concerned, and executive control refers to the monitoring and resolution of conflicts among expectations, stimuli and reactions [5]. As presented in our response to comment 1, alerting and orienting subsystems acted mainly in the information perception level, while the executive control subsystem acted mainly in the higher decision and control level. The execution control process was the most complicated, requiring the coordination of the joint operation of multiple processes and the control system to accomplish a specific goal. It would have a direct relationship with the external driving behaviors. From this viewpoint, we mentioned in our ms that 'The executive control process is a crucial sub-function of attention'. It is noted that driving contains a series of perception-decision-execution procedures, during which all three sub-functions of attention were included. The other two sub-functions are also important for the whole process, and their implication in the driving performance is worthy of study.

The manuscript was revised accordingly (please see the response to comment 1).

Comment 4: The article innovation should be presented in the Introduction.

Response 4: Thank you for your helpful comment. We revised the title of section 4.4 in the manuscript to “Novelty and limitations”, and added the following paragraph to the manuscript (Page 16, Line 566).

In this research, a unified experimental and analytical schema for multimodal data including the behavior, EEG activity and psychological performance was presented to explore the underlying cognitive mechanism of attention in driving. Through the comparative analysis of different groups of participants, the quantitative correlation between executive control function and driving performance was established, and the spatiotemporal activity of the brain during this procedure was revealed. The relationship between dangerous driving behavior and attention, along with CSIs and other parameters, was disclosed.

Comment 5: Is there an applicability of the research findings?

Response 5: Thank you for your constructive comment. About the applicability of the research findings, the following paragraph was added in the revised manuscript (Page 16, Line 573).

Based on the presented methods and the acquired results, the attentional state of the driver could be monitored through EEG signals to avoid the distraction, and the dangerous driving behaviours including errors and elapses could be prevented. Besides, the attentional and behavioural characteristics of the drivers can be analysed in advance and the subject-specific driving style can be evaluated. Accordingly, different real-time online human-computer interaction schemes can be provided for different kinds of drivers. Furthermore, the individual’s driving performance and their EEG performance could be mutually corroborated, which would supply new reference for driving training and administration. In general, the presented schema supplies a new kind of intelligent human-computer interaction method, and this active safety control would significantly improve the driving safety. Except for driving, the research findings would be applied to other life risk activities.

Comment 6: In the Discussion it would be better to have seen more use of terms like 'originality' and 'significance'. There is no clear conclusion on why the research findings are significant.

Response 6: Thank you for your critical comments. According to your comments, we have revised some expressions in the discussion (Page 13, Line 437; Page 14, Line 445; Page14, Line 454; Page 14, Line 475; Page 16, Line 552; Page 16, Line 559; Page 16, Line 562;) and added a paragraph of the article innovation in section 4.4. Besides, the following paragraph were added in the conclusion (Page 17, Line 639).

This research provided evidence of a close relationship between executive control functions and driving performance, which supplies new reference for intelligent human-computer interaction and active safety control in driving.

  1. Posner, M.I.; Petersen, S.E. The Attention System Of The Human Brain. Annual Review of Neuroscience 1990, 13, 25-42.
  2. Posner, M.I.; Rothbart, M.K. Research on Attention Networks as a Model for the Integration of Psychological Science. Annual Review of Psychology 2007, 58, 1-23, doi:http://dx.doi.org/10.1146/annurev.psych.58.110405.085516.
  3. Corbetta, M.; Shulman, G.L. Control of Goal-Directed and Stimulus-Driven Attention in the Brain. Nature Reviews Neuroscience 2002, 3, 215-229, doi:http://dx.doi.org/10.1038/nrn755
  4. Chun, M.M. A neuromarker of sustained attention from whole-brain functional connectivity. Nature Neuroscience 2016, 19, 165–171, doi:https://doi.org/10.1038/nn.4179.
  5. Petersen, S.E.; Posner, M.I. The Attention System of the Human Brain: 20 Years After. Annual Review of Neuroscience 2012, 35, 73-89.

Reviewer 2 Report

The authors have presented a very interesting work about the correlation between driving performance and executive control functions. 

The study is well written, well organized, and the results are very robust.

The authors have defined the effects of CSI and task transition types on EEG in different groups; however, The authors should do a more in-depth analysis of each group. For instance, the age, how experient they are, how long they usually drive for a week or month, and how heavy the traffic experience is for each one. This would help conclude how previous experience could change the response behavior of each driver. The authors have clarified some of these suggestions at section 4.4. However, the definition of these parameters is crucial to help consolidate a bigger picture of your work.

as a minor suggestion:
you should better describe how the clustering were done (section 3.1.1).

As a suggestion for future work, I think the authors should monitor each individual's training performance to measure their EEG performance improvement. This analysis is fascinating, not only for driving but for several other life risk activities. This result could be a very interesting product for evaluating a given professional performance.

Author Response

Response to Reviewer 2 Comments

Point 1: The authors have defined the effects of CSI and task transition types on EEG in different groups; however, the authors should do a more in-depth analysis of each group. For instance, the age, how experiment they are, how long they usually drive for a week or month, and how heavy the traffic experience is for each one. This would help conclude how previous experience could change the response behavior of each driver. The authors have clarified some of these suggestions at section 4.4. However, the definition of these parameters is crucial to help consolidate a bigger picture of your work.

Response 1: Thanks indeed for your insightful comments. As you pointed out, the demographic factors of different groups, such as the age, the educational backgrounds, the driving experience including the driving frequency and the load, might all have significant effect on the cognition and behaviour of the drivers. In our revised manuscript, we compared the genders, ages, driving years and education levels among the three groups and no significant difference was detected). However, we clarified at section 4.4 that the present study is principally limited by the relatively small sample size of the subjects. The definition of all the related parameters and a larger sample size would be crucial to help consolidate a bigger picture of our work. The individual’s driving performance and their EEG performance could be mutually corroborated, which would supply new reference for driving training and administration. But in our work, their interaction dynamics can’t be analysed because of the insufficient repeated measures of the subjects from both the cognitive and the behavioural sides. We would like to conduct a longitudinal cohort study and we believe that the very interesting and more robust results would be obtained.

The manuscript was revised accordingly (Page 6, Line 252; Page 16, Line 592) as follows.

There was no significant difference in genders, ages, driving years and education levels among the three groups (χ2 = 4.836, P = 0.089; F = 0.149, P = 0.862; F = 0.102, P = 0.903; χ2 = 2.978, P = 0.561 respectively).

Besides these factors, the other demographic factors of different groups, such as the driving experience including the driving frequency and the load, might all have significant effect on the cognition and behavior of the drivers. The definition of all the related parameters and a larger sample size would be crucial to help consolidate a bigger picture of our work. The individual’s driving performance and their EEG performance could be mutually corroborated, which would supply new reference for driving training and administration. But in our work, their interaction dynamics can’t be analysed because of the insufficient repeated measures of the subjects from both the cognitive and the behavioural sides. We would like to conduct a longitudinal cohort study and we believe that the very interesting and more robust results would be obtained.

Point 2: You should better describe how the clustering were done (section 3.1.1).

Response 2: Thanks indeed for your constructive suggestion. The following paragraph were added in the manuscript (section 3.1.1, Page 6, Line 242).

The 35 subjects were divided into different groups according to their aberrant behaviors (errors and lapses) using the systematic clustering ("bottom-up" aggregation, Euclidean distances, Shortest distance algorithm). Initially each subject belonged to the different categories. Then, the pair of subjects with the shortest distance were merged into one category. The distance between this category and the other categories were calculated and merge the two nearest categories. Continue this procedure until all the categories were merged into one. Three categories were set in advance and the subjects were classified according to the pedigree cluster diagram.

Point 3: As a suggestion for future work, I think the authors should monitor each individual's training performance to measure their EEG performance improvement. This analysis is fascinating, not only for driving but for several other life risk activities. This result could be a very interesting product for evaluating a given professional performance.

Response 3: Thank you very much for your guiding suggestions and encouragement. The individual’s driving performance and their EEG performance could be mutually corroborated, which would supply new reference for driving training and administration.

Regarding the application of the research findings, the following paragraph was added in the revised manuscript (Page 16, Line 573).

Based on the presented methods and the acquired results, the attentional state of the driver could be monitored through EEG signals to avoid the distraction, and the dangerous driving behaviours including errors and elapses could be prevented. Besides, the attentional and behavioural characteristics of the drivers can be analysed in advance and the subject-specific driving style can be evaluated. Accordingly, different real-time online human-computer interaction schemes can be provided for different kinds of drivers. Furthermore, the individual’s driving performance and their EEG performance could be mutually corroborated, which would supply new reference for driving training and administration. In general, the presented schema supplies a new kind of intelligent human-computer interaction method, and this active safety control would significantly improve the driving safety. Except for driving, the research findings would be applied to other life risk activities.